# Rumors of Psychedelics, Psychotropics and Related Derivatives in *Vachellia* and *Senegalia* in Contrast with Verified Records in Australian *Acacia*

**DOI:** 10.3390/plants11233356

**Published:** 2022-12-02

**Authors:** Nicholas J. Sadgrove

**Affiliations:** Department of Botany and Plant Biotechnology, University of Johannesburg (Auckland Park Campus), Auckland Park, P.O. Box 524, Johannesburg 2006, South Africa; nicholas.sadgrove@gmail.com

**Keywords:** dimethyltryptamine (DMT), 5-methoxydimethyltryptamine, tryptamine, phenethylamine, psychopharmacology

## Abstract

There are almost 1000 species of *Acacia sensu stricto* in Australia, while the 44 species and 4 subspecies in southern Africa were taxonomically revised in the year 2011 to *Senegalia* and *Vachellia*. There are rumors of a chemical similarity between the Australian *Acacia* and their southern African sister genera. Chemical analysis has unequivocally demonstrated the presence of tryptamines (i.e., DMT), β-carbolines, histamines, and phenethylamines in Australian species. However, reliable published data were not found in support of similar alkaloids in southern African (or even African) species, indicating the need for exploratory phytochemical analysis. Interestingly, the Australian species are more like the *Vachellia* and *Senegalia* from the Americas. While many reliable chemical studies have been found, there are several more that report only tentative results. Tentative data and anecdotal accounts are included in the current review to guide researchers to areas where further work can be done. For example, the current review encourages further phytochemical work to confirm if the two metabolite families, tryptamine and β-carboline alkaloids, occur together in a single specimen. Tryptamines and β-carbolines are the prerequisite ingredients of the South American psychotropic drink ayahuasca, which utilizes two different species to create this synergistic combination. These observations and others are discussed in light of geochemical variability, the potential ethnobotanical implications, and the need for further research to confirm or nullify anecdotal reports and tentative chromatographic/spectroscopic data in southern African species.

## 1. Introduction

In the world there are >1000 species of *Acacia sensu stricto* [1], the majority of which (975) are endemic to Australia [2]. Debates over a reclassification of the genus were ongoing for decades, but when evidence from phylogenetic analysis had mounted, it became clear that the genus is not monophyletic. By 2003 strong empirical corroboration had accumulated in favor of the proposal for the retypification that was put forward by an Australian group of botanists [3,4]. The revision was debated and eventually approved at the 17th International Botanical Congress, held in Melbourne in 2011, and the species that were previously circumscribed as *Acacia* in Africa and America were revised to two genera, namely *Senegalia* and *Vachellia* [1]. In southern Africa, this meant revision of at least 44 species and 4 subspecies, corresponding to those listed under the name of *Acacia* in the book of van Wyk and van Wyk [5] and a popular field guide of South African *Acacia* [6].

The decision to rename African *Acacia* seemed to contradict the axiom that the name should be retained where it was first used. While species of *Acacia* appear in Egyptian symbolism, and there are frequent mentions of African *Acacia* in the Christian bible, in earlier translations they are denoted by a different name [7]. The name *Acacia* was first used in the 14th century, long before the lands of Australia were opened to the rest of the world, and by the year 1747, when Linnaeus had started using *Acacia* as the name of the genus [2], botanical expeditions to Australia were still decades away. By 1754, Philip Miller formalized the name *Acacia* in his work ‘The Gardeners Dictionary’ [8], using it to describe exotic species in his home country; however, specific details of the provenance of those accessions were not found for the purpose of this review. They were likely to have been Asian because the justification put forward to retain the name in Australian biota was that references to *Acacia* prior to Miller’s work in 1754 have no formal taxonomic standing today, as it was before the official starting point of modern taxonomic nomenclature [2]. For this reason, other arguments could be considered, and retaining the name for the Australian species was considered favorable, so that fewer species would require revision.

Perhaps another incentive to retain the name in Australia has been that a species of *Acacia* is the national floral emblem [9]. This underscores the colonial Australian’s appreciation of the ‘wattles’ (vernacular for *Acacia*), namely golden wattle (*Acacia pycnantha* Benth.). The attraction of the colonialists to the golden wattle has been rumored to have been for reasons other than the aesthetic display of flowers [10], but no historic records are available to corroborate this insinuation.

The ethnobotanical use of *Acacia* by the first Australians is not documented in enough detail to ascertain if they were used for psychotropic or psychedelic experiences. Ethnobotanists and archaeologists are divided according to how this should be interpreted, with some protesting that the first Australians did not utilize the psychotropic properties of the endemic flora. However, the rituals practiced by Australian ‘Aboriginal men of high degree’ [11] were performed in strict secrecy [12], as it was knowledge afforded to initiates who had graduated to the required degree of learning [13]. Without knowledge of their secret practices, no conclusions can be drawn.

It is also possible that early Australian ethnobotanists refrained from recording or disseminating information related to psychedelic use, as there were significant Christian taboos that prohibited them from openly discussing such phenomena, similar to the fate of psychedelics in the Americas [14]. Nevertheless, by the late 1940s Australian chemists were publishing their findings of alkaloids in *Acacia*, not knowing they were psychedelics until the late 1950s when the biological activity of DMT was published under the name ‘psychotomimetic’, meaning imitator of psychosis [15].

Research into the alkaloids of *Acacia* continued into the 70s, at which time a taboo on the subject became prohibitive. Consequently, modern peer reviewed phytochemical studies of the alkaloids of *Acacia* have not been published, and details of intraspecific chemical variability are either non-existent or withheld. Where modern chemical studies have been conducted, they were commonly part of private research and details are not accessible to the wider scientific community. Findings are disseminated at ‘exclusive’ conferences or published in non-indexed magazines with a niche readership.

Furthermore, publications featuring alkaloids from the African *Senegalia* and *Vachellia* have not used acceptable methods of chemical analysis, neither today nor in the 20th century. Therefore, knowledge of the classes of compounds and their identities in these genera are tentative only, or speculative. Thus, chemical information of *Acacia sensu lato* derives from studies of Australian, American and Asian species. The dominant types of psychotropic alkaloids belong to the (a) tryptamines, (b) β-carbolines, and (c) phenethylamines (Figure 1).

This current review examines and comments on the quality of research articles describing psychedelics in *Acacia* and related genera, and the feasibility of anecdotal accounts. The research studies were identified using three databases, i.e., PubMed, Google Scholar and SciFinder. Anecdotal or non-verified sources were identified using pedestrian websites and claims made in the market (seed merchants, etc.). All sources are divided into ‘verified’ and ‘anecdotal’ (or speculative). The term ‘related genera’ is used here to infer any genera previously circumscribed as *Acacia*, and the currently excepted names were determined from Kew’s ‘World Checklist of Vascular Plants’ [16].

There are a significant number of researchers who report experimental outcomes that were generated from techniques that are unreliable and were therefore not published in peer reviewed articles. This will be explained further, and in cases of verified research, the lack of consistency across studies will be explained in light of the new paradigm of chemophenetics [17].

## 2. Australian *Acacia*

The vernacular used to denote the Australian species of *Acacia* is ‘wattle’ [9]. In the spring, wattle produces a dense display of yellow-gold flowers that are visible from a distance and can sometimes color an entire landscape where the species is dominant. The floral display is what earned the genus its ornamental value, leading to its distribution across the world, settling into gardens and arboretums. Species of Australian *Acacia* are now considered invasive in the Americas and South Africa.

Research on the psychedelic properties of *Acacia* has remained relatively inactive during the last 40 years. Academics who have written specifically about the psychedelics of *Acacia*, or of other species, are often retired from universities, or have some success and financial security (pers. observation). However, with the modern paradigm shift that recognizes the consequences of the opioid epidemic [18] and the apparent contradiction in regulatory control of substances not proven to be addictive, the school of psychedelics is being re-examined for value, particularly to fill the gaps in psychiatry [19]. While the pharmaceutical industry has demonstrated success in treating psychiatric disorders, there is a minority of non-responders to conventional drugs who may benefit from alternative therapies.

In this new paradigm, many scholars are integrating the knowledge that was accumulated in the mid-20th century, to identify research gaps and re-commence phytochemical and pharmacological studies that focus on identifying uses for the indigenous flora. The current review represents this initiative by dividing the research between verified and anecdotal, and by encouraging further work toward elucidating the chemogeography of the Australian genus *Acacia*.

Australian species are often comprised of several chemotypes, evident from studies of other species [13]. For this reason, it is expected that phytochemical reports of species in *Acacia* are geographically specific, and specimens of the same species that are collected from other locations are likely to be chemically different. Thus, the information presented in the current review is not expected to be definitive, but rather, it requires elaboration. It would be useful to have further knowledge of variation across the endemic distribution of the species.

### 2.1. Verified References of Psychedelics, Psychotropics and Related Derivatives in Australian Acacia

According to a study from CSIRO by Collins [20], 79 out of 127 species of *Acacia* have tested positive for alkaloids, but only a few of these were followed up with phytochemical analysis, and psychedelics were confirmed in just a select few of these. The records provided in Table 1 include these records as well as others that were judged to be verified if the finding was published in a peer reviewed journal and was authored by a reputable chemist who verified their work using complementary methods, such as by synthetically converting the tentatively identified product into predictable and recognizable derivatives, or by using gas chromatography with authentic standards. Only one verified source was necessary for inclusion in Table 1, and if further records were found that were tentative or speculative, they are mentioned and declared in the table as well.

The data in Table 1 indicate that several species of *Acacia* have high enough yields of DMT in leaves to validate the anecdotal claims of psychedelic effects from their extracts. However, variation in the ratio of DMT and NMT indicate that psychedelic effects vary. The data indicate that the richest confirmed source of DMT is *A. maidenii*, and other confirmed sources are generally at lower yields. Out of all the verified studies, the phenethylamines are more common, also more common in higher yields. Lastly, evidence that the β-carbolines occur together with the tryptamines in a single biota is tentative, as verified references of tryptamines are coupled with unverified references of β-carbolines in the same biota, and vice versa.

### 2.2. Rumours and Anecdotes of Psychedelics, Psychotropics and Related Derivatives in Australian Acacia

In addition to those mentioned in Table 2, there are species of *Acacia* that are known to be used with pituri, a mouth tobacco that is either made from *Duboisia hopwoodii* (F.Muell.) F.Muell. [32], or a local species of *Nicotiana* L. [33]. The species used for pituri were called *wirra* by the Aboriginal people and were used in combination with either *D. hopwoodii* or *Nicotiana*. The species called *wirra* were *Acacia aneura* F.Muell. ex Benth., *A. calcicola* Forde and Ising, *A. coriacea* DC., *A. estrophiolata* F.Muell., *A. ligulata* A.Cunn. ex Benth., *A. pruinocarpa* Tindale, *A. beauverdiana* Ewart and Sharman, and *A. salicina* Lindl. [34].

The majority of records indicate that the ash of the burnt *Acacia* was used, instead of the raw plant material [12,33]. In such cases, it is possible that the ash from these species enhanced transdermal absorption of the nicotine from the biota, as basification (ash is basic) will increase lipophilicity, helping the nicotine and nornicotine to penetrate the wall of the mouth [35]. Moreover the notion that fresh material from psychedelic biota can be used to achieve psychoactive effects is an attractive hypothetical. The combination of nicotine with psychedelic alkaloids is not new to ethnopharmacology, as it was a practice sometimes used in the traditional preparation of ayahuasca by the South Americans [36]. However, if the biota was used in a fresh form, the psychedelic effects from *N*,*N*-dimethylated tryptamines could not possibly be enacted because they are metabolized too quickly in first pass metabolism for the effects to be enacted. A counter argument could be that transdermal absorption in the oral cavity is not subjected to first pass metabolism [35], but this needs to be validated by research.

However, the notion that fresh material was mixed with pituri appears to be a misunderstanding [12]. This misunderstanding may be derived from speculation that pituri induced psychedelic effects in some cases; however, this may be due to selective use of the root of *D. hopwoodii* which yields hyoscyamine and scopolamine [12], two tropane alkaloids with anticholinergic activity capable of creating a ‘religious experience’ [37].

Unsubstantiated anecdotes of psychotropic alkaloids or psychotropic effects from Australian species of *Acacia* often derive from private studies. These ‘underground’ screenings often use modern chromatographic instruments, generating potentially reliable data that have not been published or peer reviewed. Such information is disseminated at ‘exclusive’ conferences and between colleagues, or published in niche non-indexed magazines, such as ‘The Entheogen Review’ (http://www.entheogenreview.com/, accessed on 3 November 2022), either anonymously or under a pseudonym. Anecdotes also come from thin layer chromatography (TLC) analysis, which is generally useful for exploring consistency between species with known chemistry but should not be used as a final step in chemical assignment. Co-migrating spots or bands on the TLC plate are not diagnostic but rather predictive, requiring follow-up, and the degree of predictability diminishes outside of the Rf values ranging from 0.2–0.7.

Lastly, claims of psychoactivity may come from unregulated or recreational use (human bioassay), but such cases are subjective and cannot elucidate the chemical composition, as effects can be caused by a diversity of alkaloids. Unfortunately, the taxonomic determination associated with anecdotal use is also in doubt, as the layman is not skilled at keying out or identifying *Acacia* to species level. Nevertheless, details of their findings are often reported on internet sites that are eventually taken down or updated. For this reason, a survey of these anecdotal accounts is dependent upon a retrospective search of the internet, made possible by a database called ‘Wayback Machine’ (https://web.archive.org/, accessed on 3 November 2022), which seeks to retain information that is updated or removed from the World Wide Web.

Some species that were previously known as *Acacia* were revised to a different genus. For example, *Acacia angustissima* (Mill.) Kuntze is no longer the accepted name, as it was revised to *Acaciella angustissima* (Mill.) Britton and Rose. Nevertheless, under the previous name, traces of tryptamine alkaloids were tentatively identified by using an ELISA assay [38], but the inventors of the method acknowledged that the technique can only recognise a class of compound, not its absolute identity [39]. For this reason, these findings must be regarded as tentative only.

Dialogue on the psychotropic properties of Australian species of *Acacia* experienced somewhat of a renaissance in the 1990s, at which time an ‘underground’ scene had developed. According to Graham St John, the community of unregulated users referred to the *Acacia* tea as ‘Aussiewaska’ [10] which is a pun to mean Australian ayahuasca.

In using the rare term ‘entheobotany’ in describing the South American discovery of ayahuasca, St John relays the anecdote of an individual seeking *A. maidenii* in the Northern Rivers region of NSW, to experience the effects alluded to by Collins [20], but in making a mis-determination he collects *A. obtusifolia* instead, then experiences a psychedelic experience comparable to ayahuasca. Although the chemical composition of the alkaloids from *A. obtusifolia* are not verified, it was claimed that they include the combination of β-carbolines and tryptamines (Table 2) that create a psychedelic experience lasting longer than the 30-min half-life of DMT alone [10]. Many β-carbolines inhibit the enzyme that metabolizes DMT, monoamine oxidase A [40], which increases the half-life of tryptamines and extends the psychedelic experience by up to four hours in some cases [41].

**Table 2 plants-11-03356-t002:** Australian species of *Acacia* with anecdotes of psychotropic effects or psychotropic alkaloids that have not been verified by the author or in a peer reviewed publication.

*Acacia* Taxon	Anecdote
*A. alpina* F.Muell.	Dimethyltryptamine in leaf (unregulated use, expired web source) [42].
*A. auriculiformis* A.Cunn. ex Benth.	5-MeO-DMT speculated from TLC of stem bark extract [25].
*A. beauverdiana* Ewart and Sharman	Claimed to be psychoactive in expired web source (www.bushfood.net, accessed on 3 November 2022), accessed using ‘Wayback Machine’ (https://web.archive.org/).
*A. colei* Maslin and L.A.J.Thomson	Media source claimed DMT in bark at 1.8% (not verifiable) [42].
*A. cultriformis* A.Cunn. ex G.Don	Traces of tryptamine * (by TLC) and phenethylamine in leaves [21,22,27], and 5-MeO-DMT in aerial parts speculated by TLC [25].
*A. cuthbertsonii* Luehm.	Claimed to be psychoactive in expired web source (www.bushfood.net), accessed using ‘Wayback Machine’ (https://web.archive.org/) [42].
*A. delibrata* A.Cunn. ex Benth.	Claimed to be psychoactive in expired web source (www.bushfood.net), accessed using ‘Wayback Machine’ (https://web.archive.org/) [42].
*A. falcata* Desf. (syn. *Prosopis juliflora* (Sw.) DC., var. juliflora)	Claimed to be psychoactive in expired web source (www.bushfood.net), accessed using ‘Wayback Machine’ (https://web.archive.org/) [42].
*A. hamiltoniana* Maiden (syn. *A. sieberiana*)	Claims of DMT in the leaves caused by misreading of a paper that concluded none was found [43]. Claims of psychedelic effects experienced with unregulated use [42].
*A. implexa* Benth.	Claimed to be psychoactive in the media, but no primary source found [42].
*A. macradenia* Benth.	Claimed to be psychoactive in the media, but no primary source found [42].
*A. mangium* Willd.	Claims to be psychoactive from unregulated use [42].
*A. melanoxylon* R.Br.	Claims of DMT in the bark and leaves, but no primary sources confirmed this [42].
*A. mucronata* subsp. *longifolia* (Benth.) Court	DMT, NMT, tryptamine * speculated from TLC migration patterns [44].
*A. obtusifolia* A.Cunn.	Claims of NMT, tryptamine *, harman and norharman, tentative 5-MeO-DMT from a network of anonymous authors whose works appear in ‘The Entheogen Review’ (http://www.entheogenreview.com/) [42].
*A. penninervis* Sieber ex DC.	Claimed to be psychoactive in expired web source (www.bushfood.net), accessed using ‘Wayback Machine’ (https://web.archive.org/) [42].
*A. pycnantha* Benth.	An oral presenter claimed to have identified 0.4% DMT [42].
*A. retinodes* Schltdl.	Speculation of DMT and NMT in aerial parts [42].
*A. sophorae* (Labill.) R.Br.	Alkaloids in leaves, stems and unripe seed pods [20,27] possibly tryptamine alkaloids [42].
*A. victoriae* Benth.	Speculation from TLC of DMT in aerial parts and 5-MeO-DMT in roots [25]. Unregulated users claim to have experienced psychedelic effects [42].

* All compound names are listed according to the claims of the authors.

## 3. American *Acacia*, *Senegalia* and *Vachellia*

Although the American species of *Acacia sensu lato* (*Acacia*, *Senegalia* and *Vachellia*) are potentially rich in psychedelic alkaloids, with or without β-carbolines, the ethnobotanical interest in the species is not comparable to that in Australia, as there are preferred alternatives in the Americas, such as the stem of *Banisteriopsis caapi* (Spruce ex Griseb.) Morton, for β-carbolines (harmala alkaloids) [45], or aerial parts of *Peganum harmala* L., also for harmala alkaloids [40], and leaves of *Psychotria viridis* Ruiz and Pav. [46], or root bark of *Mimosa tenuiflora* (Willd.) Poir. (syn. *M. hostilis* Benth.) [47], as a source of DMT. It is noteworthy that *M. tenuiflora* was previously named *Acacia hostilis* Mart. but has since been revised.

Because phytochemical analysis of the American species is limited, only two verifiable records have been found. However, there is much speculation about psychedelic-type alkaloids in the other species, but these are generally anecdotal or derive from inconclusive analysis of TLC migrating patterns.

### 3.1. Verified References of Psychedelics, Psychotropics and Related Derivatives in American Senegalia and Vachellia

Out of the two species in Table 3, *A. simplex* from Argentina yields 3.6% of alkaloids that consist of DMT and 2-methyl-tetrahydro-β-carboline. This is a confirmed report of these two classes of compound together. However, the hydrogenated derivatives of the β-carboline are significantly less effective at inhibition of monoamine oxidase A (MAO-A) [48], meaning it is unlikely to create the effects of ayahuasca in the strict sense. However, it is a moderate non-selective serotonin reuptake inhibitor [49], which may delay the meeting of DMT with MAO-A.

### 3.2. Rumours and Anecdotes of Psychedelics, Psychotropics and Related Derivatives in American Senegalia and Vachellia

The majority of entries in Table 4 are speculation from migrating TLC patterns. Some anecdotal records were completely unsubstantiated, such as claims of mescaline and amphetamines. Nevertheless, there are many tentative records of phenethylamines, which is feasible, due to the verified studies that make similar reports. This demonstrates the potential for a metabolomics overlap between the Australian and American species.

## 4. South African *Senegalia* and *Vachellia*

Table 5 lists several unverified claims of psychedelic compounds in African species. Rumors of psychedelics in South African species derive from studies that demonstrate psychotropic effects in animal studies, or from ethnobotanical records. Several studies were found that either misinterpreted other studies, or use thin layer chromatography, giving only tentative, albeit possibly incorrect results. Where studies were found that were tentative, they were usually focused on a specimen of either a central African or southern African species. Studies of South African species were frequently focused on biota harvested in another country, either African or in a country neighboring north-east Africa where some African species occur naturally. In such cases, the studies were merely tentative, as previously mentioned, or if by chance accurate, the biota is from a population that may be chemically different to the same species biota growing naturally in southern Africa, a consequence of chemogeography (or chemotypes). The current review encourages phytochemical analysis of the biota mentioned in Table 5, to verify the leads listed.

## 5. Description and Mechanism of Psychotropics in *Acacia*, *Senegalia* and *Vachellia*

### 5.1. The Tryptamines

The three psychotropic alkaloids in *Acacia*, *Senegalia* and *Vachellia* are NMT, DMT and 5-MeO-DMT (Figure 2). Psychedelic indole tryptamines confer psychedelic effects via the serotonergic system, by acting on the serotonin receptor, also known as the 5-hydroxytrypamine receptor (5-HT receptor) [61]. However, the neurological response by agonism of 5-HT by DMT is not the same as the effects of serotonin agonism [62] due to differences of binding affinity to the different receptor subtypes, and by the ability of serotonin to engage a β-arrestin2-mediated signaling cascade in the frontal cortex [63]. The three major recognition sites relevant to DMT are 5-HT_1_ and 5-HT_2_, but DMT binds to many other recognition sites that are further divided into subtypes. These include the 5-HT_1A_, 5-HT_1B_, 5-HT_1D_, 5-HT_2A_, 5-HT_2B_, 5-HT_2C_, 5-HT_5A_, 5-HT_6_ and 5-HT_7_ receptors, with binding affinities ranging from 39 nM to 2.1 µM [61].

The most important sites for the psychedelic effects of DMT are 5-HT_1A_, 5-HT_2A_, and 5-HT_2C_, enacting their effects in modulation according to differences in binding affinity. It was realized that affinity for 5-HT_1A_ reduces the psychedelic effects through agonism of 5-HT_2A_. Because 5-MeO-DMT has a lower affinity for 5-HT_1A_ than DMT, it confers stronger psychedelic effects [61]. In addition, agonism of 5-HT_1A_ creates no obvious behavior modification to mammals [64], so selective agonists for 5-HT_2A_ are expected to have higher potency.

Tryptamines are metabolized by the enzyme monoamine oxidase A (MAO-A), which deaminates and oxidizes the ethyl moiety to produce indole acetic acid. This metabolic process is rapid, meaning that orally administered DMT is not active unless administered concomitantly with an MAO-A inhibitor. Active routes are either intravenous or via inhalation of smoke containing DMT, giving a short-lived psychedelic experience at active concentrations, lasting approximately 5–30 min [65].

In species of *Acacia* various tryptamines have been verified or tentatively identified, including DMT, 5-MeO-DMT, NMT and tryptamine in the strict sense (Table 1 and Table 2). These structures differ according to a methoxy group (DMT vs. 5-MeO-DMT), or the number of methyl groups attached to the ethylamine. DMT and 5-MeO-DMT have two methyl groups, NMT has one, and tryptamine has no methyl groups attached. These methyl groups play a significant role in the psychedelic effects of these tryptamine derivatives. For example, NMT does not cause any significant visionary experience, and pure tryptamine is not recognized as a psychedelic. Neither of these are listed in a recent comprehensive study of psychedelics [66].

### 5.2. β-Carboline Alkaloids

The β-carboline alkaloids that have been identified in *Acacia*, *Senegalia* and *Vachellia* are harman (syn. harmane), norharman (syn. norharmane, β-carboline), 2-methyltetrahydro-β-carboline (syn. *N*-methyl-tetrahydro-β-carboline), tetrahydroharman (syn. tetrahydroharmane), and *N*-methyl-tetrahydroharman (syn. 2-methyl-tetrahydroharman(e)) (Figure 3).

The β-carboline alkaloids with the strongest research basis are harmine, harmaline, and tetrahydroharmine (Figure 3). Although they were not identified in *Acacia* or the sister genera, they provide an example of the possible pharmacological effects. There are three pharmacodynamic processes that are enacted by them, which are (1) inhibition of MAO-A [40], (2) non-selective inhibition of serotonin reuptake [67] and (3) non-selective inhibition of acetylcholinesterase [68]. These alkaloids are able to enact psychotropic effects on their own, particularly if all three are present in the oral dose, creating a synergism between these three pharmacodynamic mechanisms. At the required concentration they can create an intoxicating state in mammals [69].

As previously mentioned, MAO-A is a mitochondrial enzyme that deaminates the neurotransmitters DMT, 5-MeO-DMT, NMT, serotonin, norephinephrine, and dopamine [70], and some of the β-carboline alkaloids inhibit this enzyme. The best examples to illustrate this are the harmala alkaloids, which are named after the species they were isolated from, *Peganum harmala* L. [40]. These alkaloids are substituted derivatives of the parent structure β-carboline. When they lose double bonds on ring C (tetrahydro derivatives) the pharmacokinetic and pharmacodynamic profile changes. In particular, tetrahydro derivatives are weaker inhibitors of MAO-A, but they have a stronger affinity to 5-HT_2A_ [48]. Furthermore, tetrahydro derivatives are stronger serotonin reuptake inhibitors [49].

### 5.3. Phenethylamines

Motivation to understand the biological effects of plants that cause the ‘guajilo wobbles’ (also known as ‘limberleg’) in grazing stock animals, led to the discovery of phenethylamines in a species known in vernacular language as ‘guajilo’ (*S. berlandieri* (Benth.) Britton and Rose (sy. *A. berlandieri*) [52]. This class of compound is structurally and functionally related to the amphetamines, and while the biological effects of those derived from *Acacia* are minimally understood, it is possible they enact their effects on the sympathetic nervous system, particularly by increasing the expression of norepinephrine [71].

The phenethylamines that were identified in *Acacia* and its sister genera are phenethylamine (in the strict sense), β-methyl-phenethylamine, β-methyl-*N*-methyl-phenethylamine, *N*-methyl-phenethylamine, tyramine, *N*-methyl-tyramine, and hordenine (Figure 4). Phenethylamines are uncommonly hallucinogens, even though they tend to bind selectively to the 5-HT_2_ family. They are more commonly thought of as psychostimulants. However, not much is known about the phenethylamines from various species of *Acacia*, or the effects of combining them with tryptamines (also present in extracts from *Acacia*). What is known is that they have significantly different pharmacokinetic profiles by comparison with the illicit drug methamphetamine, which has a half-life > 10 h [72]. The long half-life of methamphetamine is attributed to the methyl group positioned at the alpha carbon on the ethyl chain (Figure 4), making it inaccessible to MAO enzymes, requiring clearance by cytochrome P450s [73].

Metabolism of phenethylamines from *Acacia* is mainly by monoamine oxidase B (MAO-B), and to a significantly lesser extent by MAO-A [74]. Because harmala alkaloids are poor inhibitors of MAO-B [40], phenethylamines are generally not capable of being substituted for tryptamines in ayahuasca preparations. Due to a half-life of 5–10 min for a range of phenethylamines that are free of methyl groups on the ethyl moiety, and their *N*-methyl analogues [75], they have not been attributed much significance in recreational use. However, *A. linearis* has been known to produce a high yield of β-methyl-phenethylamine (Table 1), which has a methyl group in the beta position of the ethyl moiety. The pharmacokinetic fate of this molecule is unknown, but it may be worthy of investigation. Although it is not the same as an amphetamine (alpha methyl group), it should nevertheless be ascertained if it has a similar half-life in human plasma.

### 5.4. Histamines

Histamines are generally not regarded as psychoactive, since they have more importance in modulation of neuropathic pain [76]. However, because the histamine (H1) receptor is part of the mechanism of euphoria with the use of MDMA [77], then the presence of histamines with phenethylamines in an extract should theoretically modulate the psychotropic effects.

The histamines are normally metabolized by MAO-B [78], but amide derivatives are unlikely to follow this pathway. Only two histamines (*N*-(2-imidazol-4-yl-ethyl)-*E*-cinnamamide and *N*-(2-imidazol-4-yl-ethyl)-deca-*E*-2, *Z*-4-dienamide) (Figure 5) have been identified in one species of *Acacia*, namely *A. longifolia* (Table 1).

## 6. Conclusions

The rumors of a chemical similarity between the Australian *Acacia* and their African sister genera *Senegalia* and *Vachellia* could not be verified by a reading of the published literature, and through anecdotal accounts. The verified chemistry of species from *Acacia sensu stricto* is comprised of a diversity of psychotropic compounds, including tryptamines (i.e., DMT), β-carbolines, histamines, and phenethylamines. This information has been gathered from studies of Australian and American biota, i.e., the sister genera *Senegalia* and *Vachellia* from the Americas are chemically very similar to Australian *Acacia*. With further research the phytochemical comparison to African genera can be made.

The pharmacodynamic effect from combinations of psychotropic compounds, present in extracts of leaves, flowers or bark, is dependent upon an orchestration of receptor subtypes, either by agonism or antagonism. For this reason, the psychotropic effects from recreational or ritual use of these species cannot be predicted with any precision. Generally, the tryptamines and phenethylamines are metabolized quickly, and oral ingestion is ineffective at creating psychotropic effects. In addition, other factors may influence the half-life, such as the presence of β-carboline alkaloids in the same extract that increase the half-life of tryptamines, or methylation of the ethyl moiety in the phenethylamines, which may reduce affinity for MAO-B.

Although β-carboline alkaloids are present in some species, the evidence that they occur together with the psychedelic tryptamines is still only tentative. However, if this phenomenon is actual, it means that the psychedelic effects can be achieved by drinking extracts, rather than smoking them. This has ethnobotanical implications, particularly where minimal knowledge of psychotropic use has been captured. Furthermore, it would be noteworthy if both tryptamines and β-carbolines are present in a single botanical entity, as this has not yet been corroborated in any species, and it means that the prerequisite ingredients of the South American psychotropic drink ayahuasca, which utilizes two different species, are present in the one entity.

## Figures and Tables

**Figure 1 plants-11-03356-f001:**
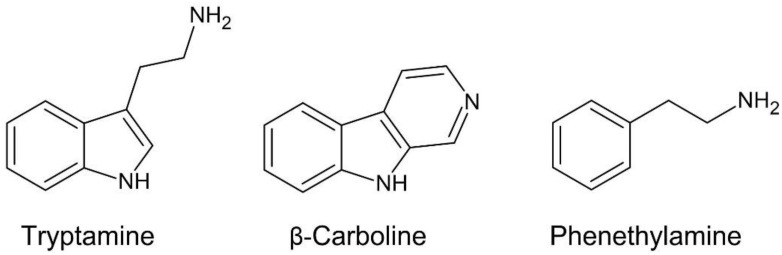
The three major classes of compounds in *Acacia*, *Senegalia* and *Vachellia*.

**Figure 2 plants-11-03356-f002:**
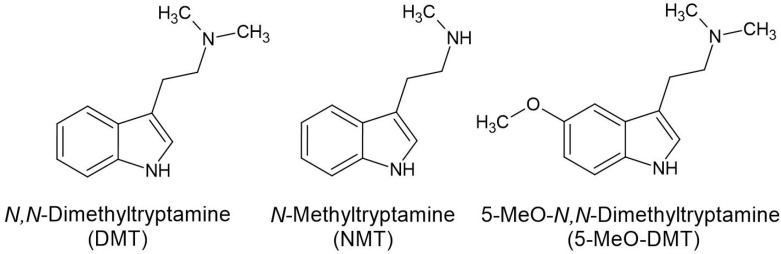
The three psychoactive tryptamines in *Acacia*, *Senegalia* and *Vachellia*.

**Figure 3 plants-11-03356-f003:**
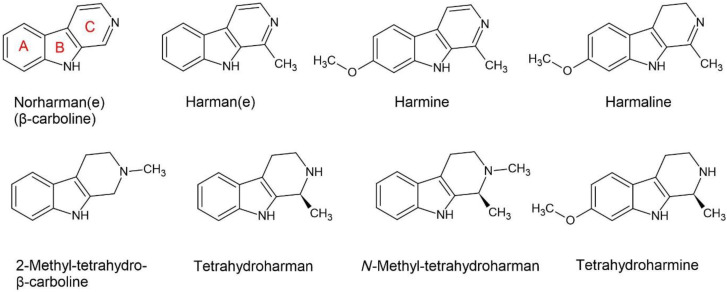
The types of β-carboline alkaloids identified in *Acacia* and its sister genera, with the exception of harmine, harmaline and tetrahydro-harmine, which are included as known pharmacophores.

**Figure 4 plants-11-03356-f004:**
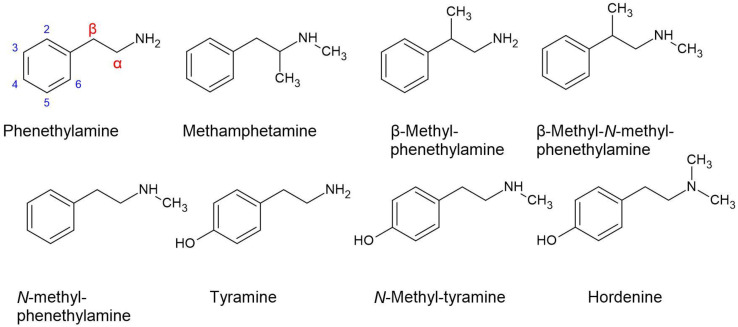
The phenethylamines (excluding methamphetamine) identified in species of *Acacia* and its sister genera.

**Figure 5 plants-11-03356-f005:**
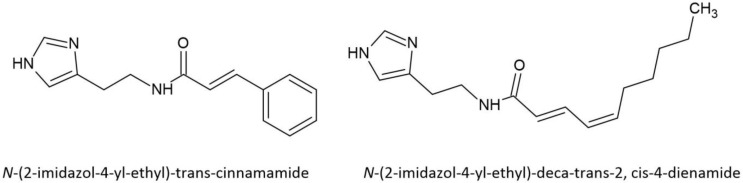
Structures of the two histamine amides identified in *Acacia longifolia*.

**Table 1 plants-11-03356-t001:** Australian species of *Acacia* with verified records of psychotropic alkaloids.

*Acacia* Taxon	Description with Reference
*A. acinacea* Lindl.	Phenethylamine * in pods (0.08%) [21].
*A. acuminata* Benth.	Tryptamine * in leaves and stems (0.7%) [22].
*A. burkittii* F.Muell. ex Benth.	Tryptamine * in leaves and stems (>1%) [22].
*A. linearifolia* J.Forbes	2.4% of β-Methyl-phenethylamine and *N*-methyl-phenethylamine from leaves [23], and 3.2% alkaloids from flowers in which most was β-methyl-phenethylamine [22].
*A. baileyana* F.Muell.	Tetrahydroharman and tryptamine * (0.02%, ratio 8:2) [24], traces of DMT speculated by TLC in seeds [25].
*A. buxifolia* A.Cunn.	0.6% phenethylamine from leaves and stems [21].
*A. cardiophylla* A.Cunn. ex Benth.	Traces of tryptamine * and phenethylamine in leaves and stems of one specimen [22], but not all specimens [20].
*A. complanata* A.Cunn. ex Benth.	*N*-methyl-tetrahydroharman (0.3%) with traces of tetrahydroharman in leaves and stem [26], unpublished claims of DMT in the bark.
*A. floribunda* (Vent.) Willd.	0.2% tryptamine * and phenethylamine in flowers [27,28]. Unpublished report of DMT in leaves (0.1%) and bark (0.5%), with traces of tryptamine, harman and norharman (unregulated community anecdote)
*A. harpophylla* F.Muell. ex Benth.	Phenethylamine * and hordenine in leaves and bark (0.1–0.6%) [20,23].
*A. holosericea* A.Cunn. ex G.Don	Hordenine (1.2%) in bark [23].
*A. kettlewelliae* Maiden	1–2% phenethylamine * in leaves and stems [22], and in another report, 1% β-methyl-phenethylamine from leaves [23].
*A. longifolia* (Andrews) Willd.	Tryptamine alkaloids (0.2–1%) with traces of phenethylamine * in aerial parts [21,27,28]. Histamine amides (*N*-(2-imidazol-4-yl-ethyl)-trans-cinnamamide and *N*-(2-imidazol-4-yl-ethyl)-deca-trans-2, cis-4-dienamide) in some specimens [29]. Claims of 0.2% DMT in the unregulated community (anecdote).
*A. maidenii* F.Muell.	NMT and DMT in leaves (0.1–0.7%) and bark [20,23]. TLC method suggested 5-MeO-DMT in woody parts and NMT in root [25].
*A. phlebophylla* F.Muell. ex H.B.Will.	DMT in leaves (0.3%) [30].
*A. podalyriifolia* A.Cunn. ex G.Don	Tryptamine * (0.1–0.3%) and phenethylamine in aerial parts [21,22,27,28].
*A. pravissima* F.Muell. ex Benth.	Phenethylamine * in leaves and stems (0.4%) [27].
*A. prominens* A.Cunn. ex G.Don	Phenethylamine * and β-methyl-phenethylamine in stems and leaves (0.2–0.7%), and flowering tops (1.8%) [21,22,27,28,31].
*A. pruinosa* A.Cunn. ex Benth.	Tryptamine * (0.02–0.1%) and traces of phenethylamine in stems, leaves, and flowers (0.4%) [27,28].
*A. spectabilis* A.Cunn. ex Benth.	Phenethylamine * in leaves and stems (0.2–0.4%) [22].
*A. suaveolens* (Sm.) Willd.	Phenethylamine * (1%) in leaves and stems [21,27].
*A. vestita* Ker Gawl.	Tryptamine * in the leaves and stem (0.1–0.3%) [22].

* All compound names are listed according to the claims of the authors.

**Table 3 plants-11-03356-t003:** Two American species, one from *Acacia* and one from *Senegalia* that have sufficient evidence of psychotropic alkaloids according to peer reviewed journal articles.

*Acacia* or *Senegalia*	Details
*A. simplex* (Sparrm.) Pedley (syn. *A. simplicifolia*)	High yield of alkaloids from leaves and stem bark (3.6%) consisting of NMT, DMT, and 2-methyl-tetrahydro-β-carboline [50,51].
*S. berlandieri* (Benth.) Britton and Rose (sy. *A. berlandieri*)	Hordenine, tyramine *, *N*-methyltyramine, *N*-methylphenethylamine in leaves [52,53,54].

* All compound names are listed according to the claims of the authors.

**Table 4 plants-11-03356-t004:** American species of *Senegalia* and *Vachellia* with tentative or speculative records of psychotropic alkaloids.

*Senegalia* or *Vachellia*	Anecdote
*S. greggii* (A.Gray) Britton and Rose (syn. *A. greggii*)	Low yield of alkaloids from leaves (0.02%), including *N*-methyl-β-methyl-phenethylamine and tyramine *, speculated from migrating TLC patterns [54].
*S. roemeriana* (Scheele) Britton and Rose (syn. *A. roemeriana*)	Low yield of alkaloids from leaves (0.4%), including β-methyl-phenethylamine, tyramine * and *N*-methyl-tyramine, speculated from migrating TLC patterns [54].
*V. aroma* (Gillies ex Hook. and Arn.) Seigler and Ebinger (syn. *A. aroma*)	Speculation of tryptamine alkaloids on the internet, claiming high yield from the seeds, but no primary source cited [42].
*V. caven* (Molina) Seigler and Ebinger (syn. *A. caven*)	Leaves combined with other substances and smoked for psychoactive effects [55]. Speculation of tryptamine alkaloids but no primary source cited [42].
*V. constricta* (Benth.) Seigler and Ebinger (syn. *A. constricta*)	Traces of β-methyl-phenethylamine, speculated from migrating TLC patterns [54].
*V. cornigera* (L.) Seigler and Ebinger (syn. *A. cornigera*)	Has been used as an aphrodisiac [55]. Speculation of tryptamines but no primary source cited [42].
*V. farnesiana* (L.) Wight and Arn. (syn. *A. farnesiana*)	Both 5-MeO-DMT and a β-carboline * were speculated by migrating TLC patterns in the extract of immature seed pods [25]. Speculation of tryptamine in stem bark, peer reviewed article [56]. Speculation of β-methyl-phenethylamine from flowers, but no primary source cited [42].
*V. rigidula* (Benth.) Seigler and Ebinger (syn. *A. rigidula*)	Low yield of alkaloids from leaves (0.3%), comprising *N*-methyl-phenethylamine and *N*-methyl-tyramine speculated from TLC migration patterns [54,57].
*V. schaffneri* (S.Watson) Seigler and Ebinger (syn. *A. schaffneri*)	Claiming β-methyl-phenethylamine, phenethylamine *, (amphetamines? and mescaline?), but no primary source identified [42].
*V. schottii* (Torr.) Seigler and Ebinger (syn. *A. schottii*)	Traces of β-methyl-phenethylamine in leaves, speculated from migrating TLC patterns [54].

* All compound names are listed according to the claims of the authors.

**Table 5 plants-11-03356-t005:** The African species of *Senegalia* and *Vachellia* that are claimed to be psychoactive or contain psychedelic alkaloids, but the evidence base is un-satisfactory.

*Senegalia* or *Vachellia*	Country	Details of Why Evidence is Insufficient
*S. laeta* (R.Br. ex Benth.) Seigler and Ebinger (syn. *A. laeta*)	Africa	Claims of DMT in the leaves caused by misreading of a paper that concluded none was found [43].
*S. mellifera* (Benth.) Seigler and Ebinger (syn. *A. mellifera*)	South Africa	Claims of DMT in the leaves caused by misreading of a paper that concluded none was found [43].
*S. polyacantha* subsp. *campylacantha* (Hochst. ex A.Rich.) Kyal. and Boatwr. (syn. *A. campylacantha*)	Africa	Old reference (1975) claims to detect traces (0.004%) of DMT in Sudanese biota, but concentration too low to be accurate with limited technology at that time [43].
*S. senegal* (L.) Britton (syn. *A. senegal*)	South African	Old reference (1975) claims to detect traces (0.003%) of DMT in Sudanese biota of the same species, but concentration too low to be accurate with limited technology at that time [43].
*V. drepanolobium* (Harms ex Y.Sjöstedt) P.J.H.Hurter (syn. *A. drepanolobium*)	East Africa	Claims of 1.4% DMT in bark and 0.5% in leaves (0.5–0.8%) by unregulated users [42].
*V. horrida* (L.) Kyal. and Boatwr. (syn. *A. horrida*)	East Africa	Claimed to be psychoactive, but no primary source found [42].
*V. karroo* (Hayne) Banfi and Galasso (syn. *A. karroo*)	South African	Roots used in Zimbabwean ethnobotany for psychoactive effects [58].
*V. nilotica* (L.) P.J.H.Hurter and Mabb. (syn. *A. nilotica*)	South African	Claims of DMT in the leaves caused by misreading of a paper that concluded none was found [43]. Speculation from TLC results of DMT [25].
*V. nilotica* subsp. *adstringens* (Schumach.) Kyal. and Boatwr. (syn. *A. nilotica* subsp. *adstringens*)	South African	Claims of DMT and harmane * in book [59], but not primary source found.
*V. oerfota* (Forssk.) Kyal. and Boatwr. (syn. *A. oerfota*)	Africa	Traces of DMT in leaves, speculated from TLC migration patterns [25].
*V. seyal* (Delile) P.J.H.Hurter (syn. *A. seyal*)	East Africa	Claims of DMT in the leaves caused by misreading of a paper that concluded none was found [43].
*V. tortilis* (Forssk.) Galasso and Banfi (syn. *A. tortilis*)	South Africa	Claims of DMT in the leaves caused by misreading of a paper that concluded none was found [43]. Another study used TLC, but with an Rf value of 0.9–0.95, but this finding is grossly inaccurate [60].

* All compound names are listed according to the claims of the authors.

## Data Availability

Not applicable.

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
