# Peer review of "Rumors of Psychedelics, Psychotropics and Related Derivatives in Vachellia and Senegalia in Contrast with Verified Records in Australian Acacia"

_plants, 2022, doi:10.3390/plants11233356_

Round 1
Reviewer 1 Report
The review, in its own words, "examines and comments on the quality of research articles, describing psychedelics in Acacia and related genera, and the feasibility of anecdotal accounts." The aim of the review is not explicitly stated, but it can be concluded that it tries to collect findings of psychedelics and some related compounds in Acacias and some related genera, and to distinguish between verified findings and speculations.
It is indeed a worthwhile task to collect findings of psychedelics (and related substances) in Acacia (and related plants), However, a review should clearly specify its scope (both in terms of plants and substances), sources and findings, and try to cover the scope as completely as possible while having clear minimum quality criteria for its sources. Unreferenced claims ("rumors") may be used to get new ideas for original research, but they are not good enough sources for a review article.
The principles of finding sources and the basis for determining "related genera" for the review is not sufficiently described. For a good review, a systematic and clearly described procedure of finding sources would be expected nowadays.
The sources were divided by the review into ‘verified’ and ‘anecdotal’, presented in separate tables. Results from thin layer chromatography were assumed to be "speculated" and were included as "anecdotal" sources.
For "anecdotal" sources, references are often unspecific or missing, therefore it is impossible to verify not only the contents but even the existence of the sources. In case if the sources are known to the researcher but confidential, it should be clearly stated. (However, this statement cannot be used to substitute missing sources nor hide data fabrication.) Otherwise, references for the sources are necessary in a review.
The scope of the substances considered in the paper is not clearly delimited. The title of the article and titles of sections 2.1, 2.2, 3.1 and 3.2 are about "psychedelics" in the respective plants. However, tables in these sections, according to their captions, list species where "psychotropic alkaloids" have been found. Furthermore, the descriptions of findings include tryptamine, phenethylamine and tyramine. It is not specified if these are the specific substances (tryptamine, phenethylamine and tyramine in itself are not "psychotropic") or their derivatives (which may be psychotropic or psychedelic, or not, depending on the specific derivative). Therefore, "psychedelics" in the title of the paper and the section titles turned into "psychotropic" in the table captions, and weakening the claim further, into compounds that are not clearly psychedelic nor psychotropic, and not even clearly defined in the findings themselves.
Classes of analytes of interest should be clearly defined for a review, and separated from each other if necessary. They cannot become fluid on the way from the title to data in order to pass non-findings for findings. In findings, it should be clear in which species which specific substances were found, including their concentrations in specific parts of the plants if available. Otherwise, the review has hardly any scientific value.
In the same line, weakly related topic of histamines seems excessive in the context of the review, and inclusion of histamines into "dominant types of psychotropic alkaloids" (page 3, lines 86-87) to justify their addition is unfounded and unacceptable. Continuing the same line, the caption of Figure 4 is "The phenethylamines identified in species of Acacia and sister genera" - and methamphetamine is displayed among others in the figure, without an indication that the unconfirmed finding has been strongly disputed. If the article aims to separate verified findings from speculative, this should be done consistently.
There are some erroneous citations in the article. On page 8, lines 225-227 mention "bark of Peganum harmala L., also for harmala alkaloids [53]" and "leaves of Mimosa tenuiflora (Willd.) Poir. (syn. M. hostilis Benth) [59], as a source of DMT". This seemed suspicious because Peganum harmala is a herbaceous plant for which "bark" does not make sense. In Mimosa tenuiflora, not leaves but root bark (or less often stem bark) is usually used for DMT. When I compared the claims to their cited sources, it became evident that the plant parts were wrongly cited. For P. harmala, the cited source (https://www.sciencedirect.com/science/article/pii/S0278691509006012?via%3Dihub#tbl1) had examined various parts of the plant but, as expected, it did not even mention bark. For Mimosa tenuiflora, as expected, only roots, not leaves had been examined in the cited source (https://chemistry.mdma.ch/hiveboard/picproxie_docs/000432467-indole-alkaloids.pdf). I did not have time to check other citations in the paper, but it is not good that both citations I happened to check turned out to be wrong.
The paper contains excessive self-citations. The single author of the paper is an author in 17 references out of 89. Not a single one of these self-citations are about Acacias or psychedelics. Among these, in 14 self-citations he is the first author of the article. There is no adequate reason for such massive self-citation.
Speculation about presence of beta-carbolines and tryptamines in a single species is too strongly stressed in the abstract and in the conclusions of the paper, as the results remain inconclusive in this respect. As the author himself tells, "verified references of tryptamines are coupled with unverified references of β-carbolines 147 in the same biota, and vice a versa". The article does not add substantial new information to present knowledge: yes, it would be fascinating if these two classes of substances were found in a single specimen, this may possibly occur, but we still do not know for sure if it does happen or not. If the author wants to concentrate his efforts on this question, it should be listed among the aims of the paper, and specific candidate species with respective specific candidate substances should be clearly presented in the conclusions so further analytical studies could proceed from these hypotheses.
In conclusion, the article needs major revision before it can be considered acceptable for publication. The task may be too big for one person without previous extensive experience in research of these specific compounds in these specific plants. A good review requires in-depth knowledge of the subject that can be obtained by long-time dedicated work. It is not so easy for someone who never worked and published on a research topic to write a good review about it. I suggest to the author collaboration with experienced researchers in the specific field who deeply know the specific plants and their uses, the specific substances and the analytical methods.
Author Response
Dear Reviewer
Thank you very much for your comprehensive comments on the quality of this manuscript. I have incorporated you comments, hopefully to your satisfaction.
Points were addressed as follows:
(The aim of the review is not explicitly stated)
I have revised the abstract to convey more clearly the objective of the current review.
An important focus of the review is to summarise anecdotal sources, to identify them as unreliable and encourage a) further research (by providing research leads), and b) to discourage the miscarriage of information in the literature, as papers continue to be published with wrong information.
My specific objective in writing this review is to follow up with phytochemical studies of southern African Vachellia and Senegalia.
(The principles of finding sources and the basis for determining "related genera" for the review is not sufficiently described.)
Details are now provided at the end of the introduction to convey that normal databases were searched, such as google scholar, PubMed and Scifinder. Regarding related genera, this terms simply means genera that were previously circumscribed as Acacia.
(For "anecdotal" sources, references are often unspecific or missing.)
The anecdotal sources were either web-based, or from incorrect information in the published literature. The main source of anecdotes came from a Wikipedia article, which is now cited in the manuscript. Other anecdotes derived from outdated websites, such as nurseries or seed merchants.
(The title of the article and titles of sections 2.1, 2.2, 3.1 and 3.2 are about "psychedelics" in the respective plants.)
The title of these sections, and the title of the review, have now been updated to read ‘psychedelics, psychotropics and related derivatives’
(the descriptions of findings include tryptamine, phenethylamine and tyramine.)
I have revised the tables to convey that the names are consistent with the presentation of the authors of the reports. I believe that ‘tryptamine’ does indeed mean the simple tryptamine, and so on. I believe this information is useful because of a close biosynthetic relationship to a compound, such as DMT.
(In the same line, weakly related topic of histamines seems excessive in the context of the review, and inclusion of histamines into "dominant types of psychotropic alkaloids" (page 3, lines 86-87) to justify their addition is unfounded and unacceptable.)
Yes, very true. I have revised Figure 1, removed histamine, and I have reduced the dominant types of psychotropic alkaloids to just three.
(Continuing the same line, the caption of Figure 4 is "The phenethylamines identified in species of Acacia and sister genera" - and methamphetamine is displayed among others in the figure, without an indication that the unconfirmed finding has been strongly disputed.)
The Figure 4 caption needed to be corrected to avoid giving the wrong impression. The inclusion of methamphetamine in Figure 4 was initially to convey the structural relationship and to aid in visualization in the section that discussed pharmacokinetics.
(In Mimosa tenuiflora, not leaves but root bark (or less often stem bark) is usually used for DMT)
Thank you for spotting this, it was an accidental cross over, putting leaves instead of bark for one species and bark instead of leaves for the other. Now corrected.
(The paper contains excessive self-citations.)
Thankfully I recently published a review on chemophenetics, so I was able to drop many of the self-citations in place for a broader coverage of the concept. I was a little disappointed to drop the citations I used to prove the intra-specific chemical diversity of Australian flora, but realized that a simple chapter would suffice.
(Speculation about presence of beta-carbolines and tryptamines in a single species is too strongly stressed in the abstract.)
I have reworded the abstract to convey that it requires further research, but I also encourage further research in this area.
Reviewer 2 Report
An interesting, well-documented, well-written review discussing the available information on the psychedelics in Vachellia and Senegalia (i.e. ex-Acacia) species.
Author Response
thank you
Round 2
Reviewer 1 Report
The article has somewhat improved due to revision. However, some problems remain. I did not check the whole article, I only bring examples of what I noticed but the whole article has to be checked either by the author or an editor.
In keywords, instead of "phenylethylamine", I suggest "phenethylamine": it would be consistent with the tables and useful to avoid confusion with a different compound, 1-phenylethylamine. Also, "methyltryptamine" is included in keywords but does not appear anywhere in the article.
Spelling errors: "merchents", "peole", "Availabe". Grammar (lines 255-257): "However, since it is a moderate non-selective serotonin reuptake inhibitor [49], which may delay the meeting of DMT with MAO-A." - here either "since" is excessive in the sentence, or a part of the sentence is missing.
The whole article needs to be checked for spelling and grammar, it is not a reviewer's task.
References [50] and [76] are incomplete. See https://www.sciencedirect.com/science/article/abs/pii/S0031942200888913 and https://pubmed.ncbi.nlm.nih.gov/15228154/
I think the article has some value but it still needs some edition before publication.
Author Response
Thank you for your encouragement, and for spotting the grammatical errors and spelling mistakes.
As requested I gave the manuscript a proof read and identified several other errors, correcting them as I saw them. There were other sentences that were incomplete, that are now grammatically correct with revision.
The keywords have been corrected according to the errors you have identified.
The suggested references were incorporated. I note that the work you have alerted me to, on A. simplicifolia, used NMR to determine their chemical assignment. This captured my attention.
Once again, thank you for taking the time to help me improve this manuscript.